# S-N Curve Characterisation for Composite Materials and Prediction of Remaining Fatigue Life Using Damage Function

**Ho Sung Kim *** and **Saijie Huang**

Discipline of Aerospace, Mechanical and Mechatronics Engineering, School of Engineering, College of Engineering, Science and Environment, The University of Newcastle, Callaghan, NSW 2308, Australia; Saijie.Huang@uon.edu.au

\* Correspondence: ho-sung.kim@newcastle.edu.au

**Abstract:** S-N curve characterisation and prediction of remaining fatigue life are studied using polyethylene terephthalate glycol-modified (PETG). A new simple method for finding a data point at the lowest number of cycles for the Kim and Zhang S-N curve model is proposed to avoid the arbitrary choice of loading rate for tensile testing. It was demonstrated that the arbitrary choice of loading rate may likely lead to an erroneous characterisation for the prediction of the remaining fatigue life. The previously proposed theoretical method for predicting the remaining fatigue life of composite materials involving the damage function was verified at a stress ratio of 0.4 for the first time. Both high to low and low to high loadings were conducted for predicting the remaining fatigue lives and a good agreement between predictions and experimental results was found. Fatigue damage consisting of cracks and whitening is described.

**Keywords:** S–N curve; fatigue damage; damage function; remaining fatigue life; PETG

## 1. Introduction

S-N curve characterisation is important not only for engineering materials but also for the fail-safe design and fatigue life prediction of various components subjected to dynamic loading. The S-N curve represented by a model [1] may be efficiently described in a diagram for applied stress (S) versus number of loading cycles (*N*). The S-N curve behaviour has been a backbone of fatigue life studies since the 19th century [2,3]. The literature shows that a data point at the lowest fatigue life for an S-N curve has been arbitrarily determined for fatigue characterisation (e.g., logN $\approx$ 2.7 [4] or 4.6 [5]), and that an ultimate strength obtained from the static test at an arbitrary loading rate (e.g., 1 mm/min [6] or 5 mm/min [7]) was used as the peak stress ($\sigma_{max}$) at the lowest number of loading cycles. However, no one seems to have paid attention to how valid such adopted ultimate strength values are when used for the fatigue characterisation of polymeric matrix materials for composites.

Eskandari and Kim [8] recently rationalised that the lowest number of loading cycles for S-N behaviour should be 0.5 in the case of a stress ratio (*R*) of zero for predicting the remaining fatigue life according to the fatigue damage theory. The difference between 1 and 0.5 cycles may be substantial on a logarithmic scale for the location factor on the S-N plane. The arbitrary choice of the lowest number of loading cycles with its corresponding static ultimate strength, thus, may lead to potentially serious errors. In addition, the prediction of fatigue life under various conditions becomes more and more complex and hence errors accumulate when the number of independent variables (e.g., applied peak stress and stress ratio) for the prediction increases. For example, when one predicts an S-N curve for a different stress ratio, the accuracy of the prediction depends on individual accuracies in both the S-N curve model and constant fatigue life (CFL) model [9,10].

Various stress ratios for fatigue are possible due to the loadings such as tension-tension (T-T) for $0 < R < 1$, tension–compression for $\chi < R < 0$, compression–tension for $\pm\infty < R < \chi$,

and compression–compression for $\pm\infty < R < 1$, where $\chi$ is the critical stress ratio [10] dependent on compressive and tensile strength ratio. Eskandari and Kim [8] proposed a framework for validation of a fatigue damage function, and a theory for predicting the remaining fatigue life at various applied stress levels with a constant $R$. They verified the theory experimentally but only at $R = 0.0$. This verification may be for a special case under the T-T loading. Additionally, it adopted fatigue data obtained from the literature [11] at an arbitrary loading rate of 1.27 mm/min with a loading frequency of 10 Hz. As such, there has been a demand to find out about what ultimate strength should be used for fatigue behaviour, and for further verification of the theory at a different stress ratio other than a stress ratio of zero using an adequately obtained ultimate strength.

On the other hand, material properties are affected by the manufacturing technology. One of the manufacturing methods brought to our attention is the 3D printing of polyethylene terephthalate glycol-modified (PETG). The literature about S-N fatigue for 3D printed PETG seems scarce. Dolzyk and Jung [7] attempted to investigate the raster orientation effect on S-N curve behaviour, however, some characteristics seemed to be obscured due to an insufficient number of data points and an invalid S-N curve model. This suggests that more experimental fatigue data with a valid model may be beneficial.

In the light of the deficiencies in the past methodology and verification for the theory of fatigue damage, the purpose of this paper was to: (a) develop a method for determining the initial peak stress within the 1st loading cycle using tensile test results; (b) verify the damage function proposed by Eskandari and Kim [8] for predicting the remaining fatigue life at a high stress ratio of 0.4 using validly determined initial peak stress; and (c) investigate the S–N fatigue behaviour of PETG.

## 2. The Theory

### 2.1. S-N Curve Model

The S-N curve model of Kim and Zhang [1,10] has been evaluated to be best suited, not only for characterisation but also for the prediction of stress ratio effect on the fatigue lives of composite materials. The number of cycles at failure ($N = N_f$) in the model with the half cycle ($N = N_0$) is given as a function of applied peak stress ($\sigma_{max}$):

$$N_f = \frac{(\sigma_{uT})^{-\beta}}{\alpha(\beta - 1)}\left[\left(\frac{\sigma_{max}}{\sigma_{uT}}\right)^{1-\beta} - 1\right] + N_0 \tag{1}$$

or inversely,

$$\sigma_{max} = \sigma_{uT}\left(\frac{\alpha(\beta - 1)\left(N_f - N_0\right)}{(\sigma_{uT})^{-\beta}} + 1\right)^{\frac{1}{1-\beta}} \tag{2}$$

where $\sigma_{uT}$ = ultimate tensile strength, and $\alpha$, $\beta$ = damage parameters.

The parameters ($\alpha$, $\beta$) are obtained from the fatigue damage rate for T–T loading and given in:

$$\frac{\partial D_f}{\partial N_f} = \alpha(\sigma_{max})^{\beta} \tag{3}$$

where $D_f$ is the fatigue damage at tensile fatigue failure [8] defined as,

$$D_f = 1 - \frac{\sigma_{max}}{\sigma_{uT}}. \tag{4}$$

A Matlab script for determining $\alpha$ and $\beta$ is given in the Appendix A.

### 2.2. Prediction of Remaining Fatigue Life

The remaining fatigue lives of composite materials when subjected to a changed $\sigma_{max}$ at a constant stress ratio ($R$) can be predicted using the fatigue damage function ($D$) for any point on an S-N plane [8,12,13]:

$$D = D_f d_f^n \tag{5}$$

where $n$ is an exponent to be determined according to the procedure described in the next section, and $d_f$ is the (general) location factor for a point on an S-N plane at an arbitrary number of cycles ($N$) and peak stress ($\sigma_{max}$), defined as:

$$d_f = \frac{\log N + 0.3}{\log N_f + 0.3} \tag{6}$$

for $N_0 = 0.5$ cycles. Note the value of 0.3 is from $-\log(N_0)$.

When a first peak stress $\sigma_{max1}$ is changed during loading to a new $\sigma_{max2}$, an iso-damage point at the new $\sigma_{max2}$ with $N = N_2$ can be identified using the location factor ($d_{f2}$) corresponding to $\sigma_{max2}$,

$$d_{f2} = d_{f1} \left( \frac{D_{f2}}{D_{f1}} \right)^{1/n} \tag{7}$$

where subscripts 1, 2 = first and second in the loading sequence respectively;

$$d_{f1} = (\log N_1 + 0.3) / \left( \log N_f + 0.3 \right) \text{ at } N = N_1 \text{ and } \sigma_{max} = \sigma_{max1};$$
$$d_{f2} = (\log N_2 + 0.3) / \left( \log N_{f2} + 0.3 \right) \text{ at } N = N_2 \text{ and } \sigma_{max} = \sigma_{max2}.$$

Subsequently, the remaining fatigue life (= $N_{f2} - N_2$) can be predicted.

### 2.3. Determination of the Exponent n

An approximately valid exponent $n$ in Equations (5) and (7) can be found according to the procedure outlined in Figure 1a with notation in Figure 1b. The procedure starts with an arbitrarily nominated initial value for $n$ (e.g., $n = 1$) and follows the two sets of calculation steps:

Step a1: $D_f$ at points B (= $D_{fB}$) and A (= $D_{fA}$) (Figure 1a) and for $\sigma_{Hmax}$ and $\sigma_{Lmax}$ respectively using Equation (4);

Step a2: $d_f$ at point b (= $d_{fb}$) using Equation (6) (i.e., $d_{fb} = \left( \frac{D_{fB}}{D_{fA}} \right)^{1/n}$);

Step b1: $\text{Log}N_f$ at point B (= $N_{HB}$) and point C (= $N_{HC}$) using Equation (1);

Step b2: $d_f$ for point C at $\sigma_{Lmax}$ (= $d_{fC} = \frac{\log N_{fB} + 0.3}{\log N_{fC} + 0.3}$) where $N_{fB}$ and $N_{fC}$ are $N_f$ (see Equation (1)) at $\sigma_{Hmax}$ and $\sigma_{Lmax}$ respectively;

Step 3: $\Delta d_f = d_{fb} - d_{fC}$

This procedure is repeated until a calculated value (= $\Delta d_f$) becomes positive for all other high ($\sigma_{Hmax}$) and low ($\sigma_{Lmax}$) stresses. If $\Delta d_f$ turns out to be negative, $n$ may be increased by typically by 0.1 or less. Then, it is repeated for other pair of stresses (i.e., $\sigma_{Hmax}$ and $\sigma_{Lmax}$). The interval (= $\sigma_{Hmax} - \sigma_{Lmax}$) may be typically 1 MPa or smaller. Finally, it is ensured $\Delta d_f$ is positive for the peak stresses. A Matlab script based on the procedure for finding a valid exponent $n$ is given in the Appendix A.

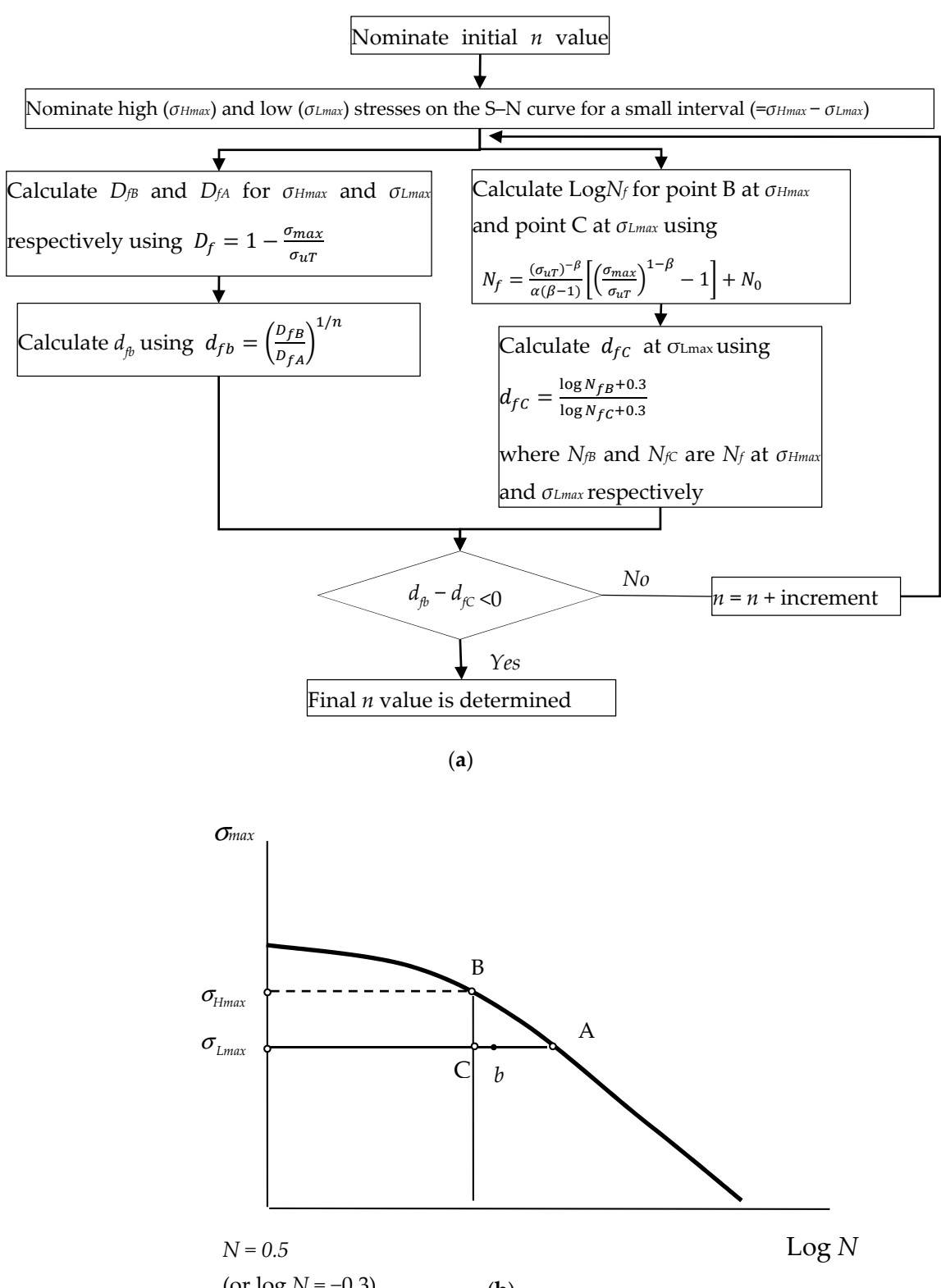

**Figure 1.** Calculation and notation: (**a**) sequence for finding exponent $n$ value at given high ($\sigma_{Hmax}$) and low ($\sigma_{Lmax}$) stresses; and (**b**) notation on schematic S–N plane.

## 3. Experimental

### 3.1. Material and Specimens

The specimen material for both tensile and fatigue testing was polyethylene terephthalate glycol-modified (PETG) supplied by PUSH PLASTIC (https://www.pushplastic.com/collections/all-filament, accessed on 5 March 2021) in the form of a filament with a diameter of 2.85 mm suitable for a 3D printer.

Dimensions and shape for both tensile and fatigue specimens were adopted from ASTM D638—Standard Test Method for Tensile Properties of Plastics as shown in Figure 2.

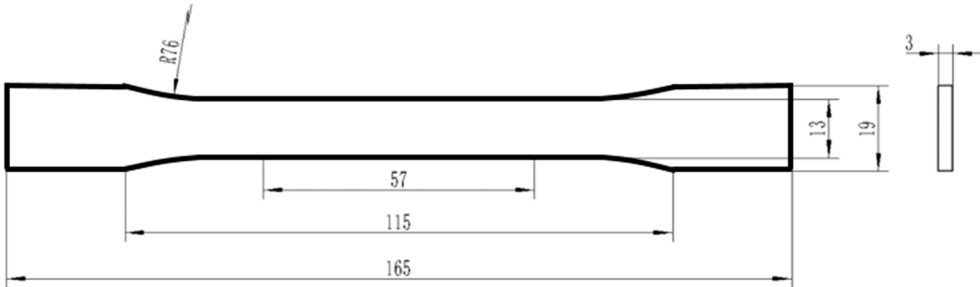

**Figure 2.** Specimen dimensions for tensile and fatigue tests.

Specimens were manufactured using a CreatBot 3D printer (Model, DX + 03; Build Volume, 300 × 250 × 520 mm (SuWei Inc., Singapore) (https://www.creatbot.com/en/creatbot-dx.html, accessed on 5 March 2021) with: nozzle temperature = 260 °C; bed temperature = 75 °C; nozzle speed = 40 mm/sec; and nozzle diameter = 0.6 mm. The raster orientation was chosen to be longitudinal, and the orientation of each specimen with respect to the manufacturing bed was set to be flatwise to avoid unwanted failure positions caused by rough surfaces near the round parts of the specimen when manufactured edgewise. A specimen thickness of 3 mm was achieved with 15 layers (= 3/0.2) by setting each layer height to 0.2 mm. Further set up details are as follows: extrusion width = 0.8 mm; infill speed = 100%; perimeters = 8, consisting of 8 top (upper) layers and 8 bottom (lower) layers; and thickness of each layer = 0.8 mm (=13/8/2) for 13 mm in gauge width of specimen.

### 3.2. Mechanical Tests

The tensile test was conducted on a universal testing machine (Shimadzu 50 kN) with a clip-on extensometer (Epsolin Model 3542, with a gauge length of 25 mm) at approximately 20 °C.

The fatigue test was conducted on a servo-hydraulic fatigue testing machine (BISS, 25 kN, http://www.biss.in/nano-plug.php, accessed on 5 March 2021) at room temperature. Fatigue loading was sinusoidal at 5 Hz and a stress ratio (*R*) was set to 0.4

## 4. Development of Method for Data Point at the Lowest Number of Loading Cycles

As mentioned at the outset, the ultimate strength ($\sigma_u$) for the data point at the lowest number of loading cycles of the S-N curve should match with that of fatigue loading rate. To this end, it was assumed that materials break at the peak stress of the fatigue load as detailed in [8]. Time to reach the peak stress ($\sigma_{max}$) from the initial valley stress ($\sigma_{min}$) of zero, accordingly, can be used as time to reach the breaking point as schematically shown in Figure 3. It may be noted that Figure 3 is a representation of cyclic loading for *R* = 0.4, in which the early part of the curve with the first cycle is idealised, given that, in reality, it may be difficult to have such a cyclic curve from a fatigue testing because of the responding time lag of the hydraulic actuator prior to reaching a set load range. An example obtained from one of fatigue specimens for $\sigma_{max}$ = 50 MPa and $\sigma_{min}$ = 20 MPa is shown in Figure 4. The breaking point within the first cycle, for this reason, should be obtained from a universal testing machine. Additionally, the cycle at the breaking point

(Figure 3) is still not exactly 0.5 and the first half cycle is inevitably slightly different from other regular half cycles because the stress ratio of the first half cycle is always meant to be zero for any first cycle T-T loading, although the other half of the first cycle is not different from the other regular half cycles. Nonetheless, it may be reasonable to approximate the cycle at the first breaking point to be a 0.5 cycle for the S-N fatigue characterisation. The corresponding time to 0.5 cycles, thus, is calculated to be 0.1 s (= $1/(5 \times 2)$) for the current 5 Hz loading frequency. A universal testing machine may be used by setting an appropriate crosshead speed if we can find the breaking stress at 0.1 sec. However, if the highest crosshead speed of a regular universal testing machine is 1000 mm/min, it may not be sufficiently high in most cases. Even if it were sufficiently high, it would be difficult with one specimen to find the matching breaking point, requiring a set of multiple specimens for a possible regression analysis as follows.

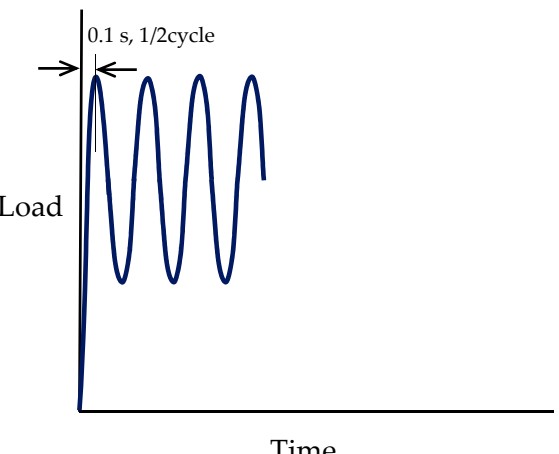

**Figure 3.** Schematic breaking point of $R = 0.4$ at 0.5 cycle.

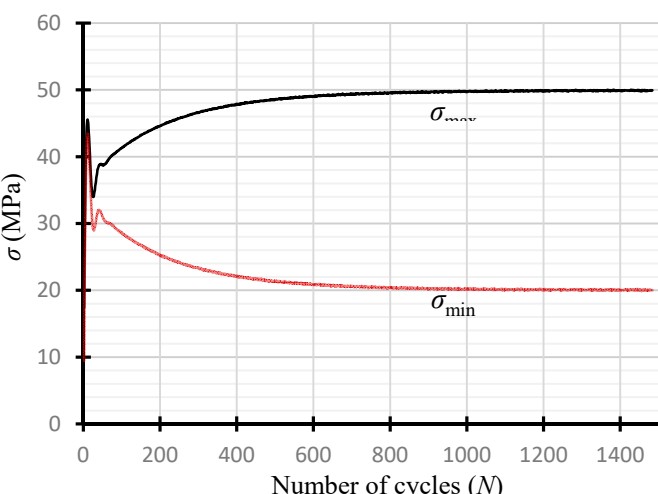

**Figure 4.** Partial results following the initial fatigue loading set-up for $\sigma_{max}$ = 50 MPa and $\sigma_{min}$ = 20 MPa.

Figure 5 shows data points obtained from multiple specimens at different crosshead speeds and it appears to be linear on stress versus log (time) with the least square line:

$$\sigma = -1.8857 \ \log(\text{time}) + 50.118 \tag{8}$$

with a Pearson's correlation coefficient ($r^2$) of 0.99. Accordingly, the extrapolated value using the least square line for the first 0.5 cycle stress ($\sigma$) at log($-1$) or 0.1 s is found to be 52 MPa. It should be noted that the extrapolated stress (52 MPa) and highest experimental

stress (50.85 MPa) obtained at a crosshead speed of 1000 mm/min are 14% and 11% higher, respectively, than the lowest (45.63 MPa) experimental ultimate stress obtained at 1 mm/min.

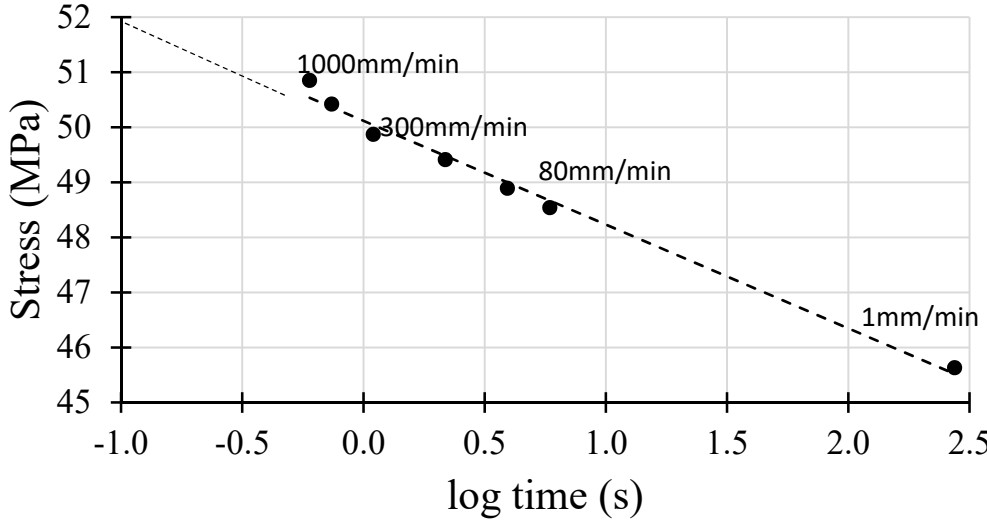

**Figure 5.** Tensile test results for extrapolation at 0.1 s (or log(−1)) to find an equivalent $\sigma_{max}$ at 0.5 cycle failure.

The tensile stress-strain curves obtained at different crosshead speeds are shown in Figure 6. Elastic moduli at 1 and 80 mm/min were measured to be both 1556 MPa and the lowest elastic modulus was measured to be 1333 MPa at a crosshead speed of 1000 mm/min. As such, they appear not to be much affected by the crosshead speed. However, the ultimate strength (or the highest stress) is seen to increase with increasing crosshead speed.

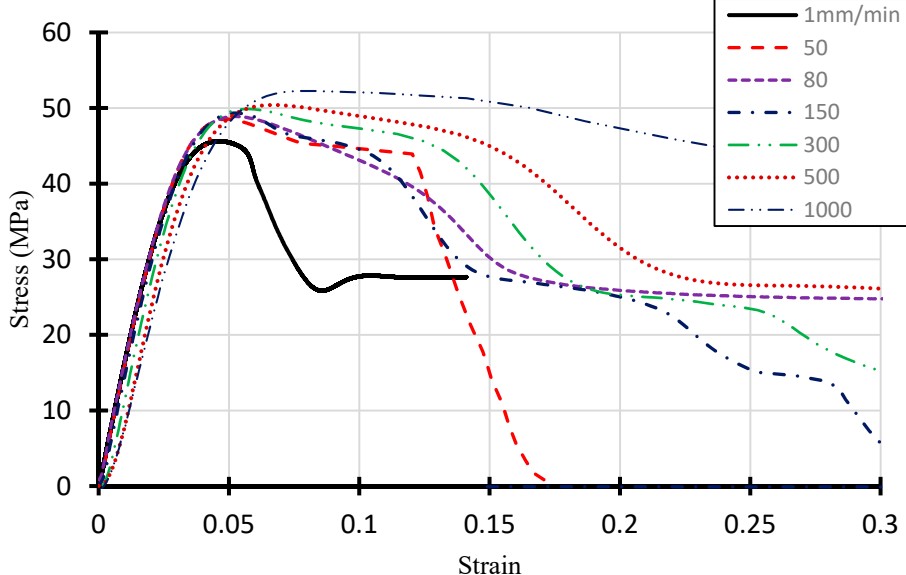

**Figure 6.** Tensile stress-strain curves obtained at a range of 1–1000 mm/min.

Figure 7 shows the failure modes of the specimens following static tensile testing. All the specimens, except one (at 1 mm/min), seem to be in a similar mode in which the fracture angle has a tendency towards 45° with respect to the loading direction, although the fracture angles are less than 45° probably due to anisotropy from the raster orientation of the specimens. The one at the lowest crosshead speed of 1 mm/min displays the material drawing which starts from the maximum stress.

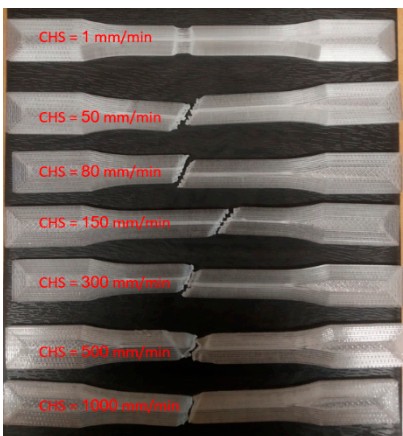

**Figure 7.** Failure modes of the specimens following tensile testing for a crosshead speed range of 1–1000 mm/min.

## 5. Fatigue Results and Discussion

Experimental data points obtained from the fatigue tests are plotted in Figure 8 with fitted S-N curves represented by Equation (1). The solid S-N curve is shown for $\sigma_u$ = 52 MPa obtained from extrapolation, and the dashed S-N curve for $\sigma_u$ = 45.6 MPa obtained at a crosshead speed of 1 mm/min. The solid S-N curve appears to represent experimental data adequately, whereas the other appears to significantly deviate from the experimental data. The deviation of the dashed line is obviously caused by the introduction of the data point for $\sigma_u$ = 45.6 MPa, which is the violation of the fatigue damage axiom [8]. The parameters $\alpha$ and $\beta$ in Equation (1) were obtained to be $\log\alpha$ = −7.016 (or $\alpha$ = 9.65 × 10$^{-8}$) and $\beta$ = 1.031 for the solid S–N curve, and $\log\alpha$ = −6.7072 (or $\alpha$ = 1.96 × 10$^{-7}$ and $\beta$ = 0.8598 for the dashed S-N curve. The parameter values for the latter would not have been possible to obtained without removing an invalid data point for $\log\left(\partial D_f/\partial N_f\right)$ versus $\log\sigma_{max}$ (see Equation (3)) from the violation resulted. Accordingly, as the results indicated, the data point for the ultimate strength corresponding to the lowest number of loading cycles should be obtained using the adequately verified method rather than at an arbitrary loading rate. Otherwise, the result is erroneous outcomes not only for the S–N characterisation but also for various predictions.

Figure 9 shows the broken fatigue specimens after fatigue testing with reference to an untested specimen (Figure 9f). The vertical straight lines in all specimens were formed longitudinally due to the moving direction of the injection nozzle of the 3D printer. The horizontal lines other than cracks in all specimens may be crazes on the surfaces following manufacturing. It is seen that the specimen with $\sigma_{max}$ = 50 MPa displays no visible cracks outside the fracture surface, indicating that fatigue damage is not much spread across the whole specimen prior to the breaking point. This is not unexpected at a high $\sigma_{max}$, because the fatigue damage tends to accumulate less at the high $\sigma_{max}$ due to the small number of loading cycles. However, all other fatigue specimens display multiple cracks in the form of damage (indicated with solid arrows) except at $\sigma_{max}$ = 35 MPa. Additionally, some whitening is seen at $\sigma_{max}$ = 43 MPa in another form of fatigue damage. It is also seen at $\sigma_{max}$ = 50 MPa that there is a tendency of the fracture angle towards 45°, resembling the broken tensile specimens (Figure 7), and a significant permanent deformation along the fracture path occurred whereas all other specimens display approximately horizontal cracking paths. The damage mechanism of the fatigued specimens mainly involves crack initiation and then propagation. The observations here would represent the final stage of the mechanism involved.

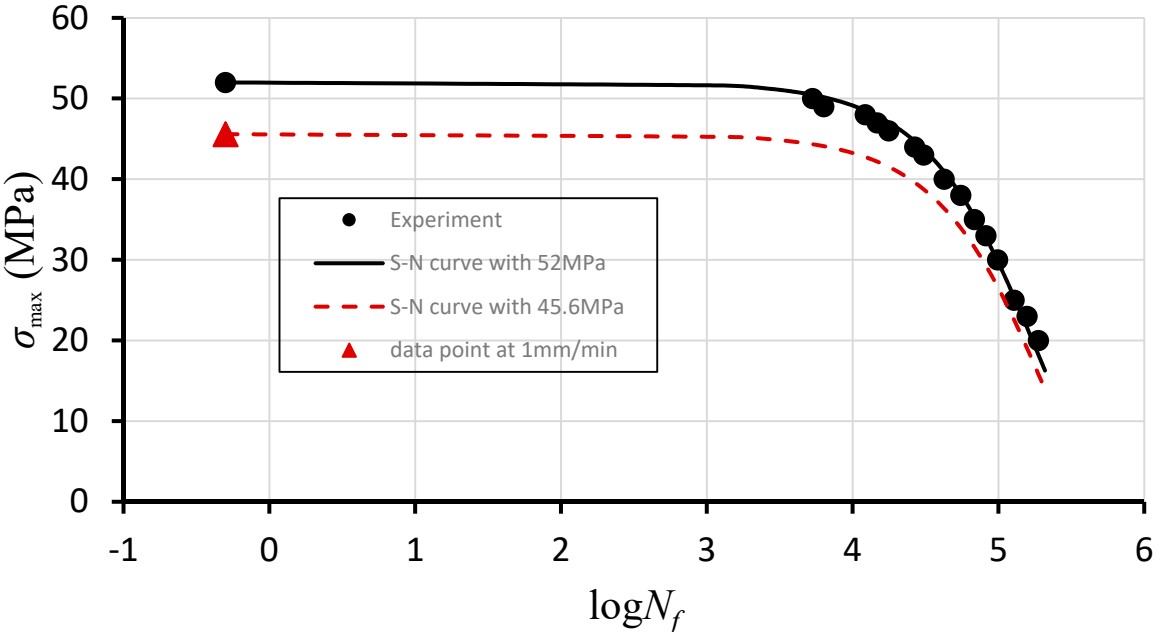

**Figure 8.** S-N curves fitted and experimental data: log $\alpha$ = −7.016 and $\beta$ = 1.031 for solid line with $\sigma_u$ = 52.0 MPa obtained from the extrapolation; and log $\alpha$ = −6.707 and $\beta$ = 0.856 for dashed line with $\sigma_u$ = 45.6 MPa obtained at a crosshead speed of 1 mm/min.

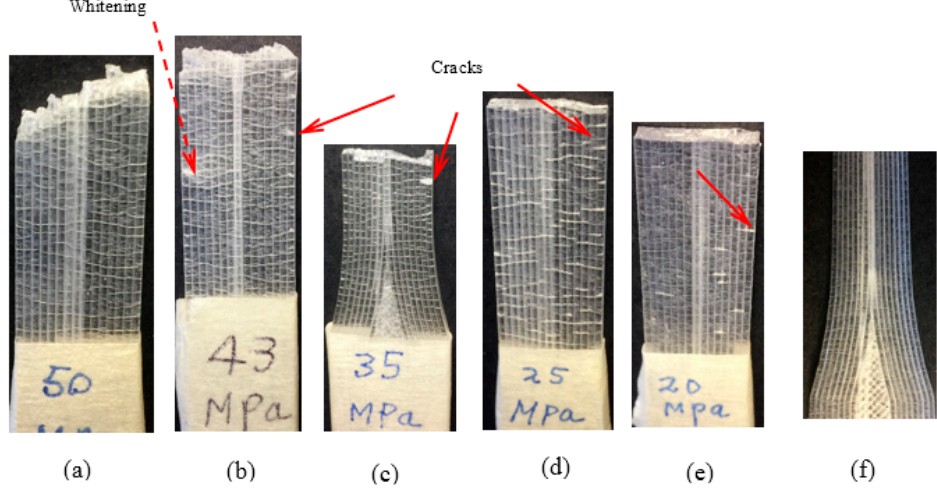

**Figure 9.** Broken fatigue specimens after fatigue testing: (**a**) $\sigma_{max}$= 50 MPa—no visible cracks outside fracture surface; (**b**) $\sigma_{max}$ = 43 MPa—multiple small cracks with whitening along the edges; (**c**) $\sigma_{max}$ = 35 MPa—one crack is clearly visible; (**d**) $\sigma_{max}$ = 25 MPa—many cracks are seen; (**e**) $\sigma_{max}$ = 20 MPa—multiple crack are visible; and (**f**) untested.

The predictions of remaining fatigue life following the change (i.e., high to low or low to high) in fatigue loading are given in Figure 10 with the S-N curve for $\sigma_u$ = 52 MPa obtained from extrapolation, experimental results, and loading paths. Each of the triangle symbols in Figure 10a,b represent the final breaking point at $N = N_{f2}$ of each fatigue specimen (notation is given using one of the data sets in the figure.) The accuracies of the prediction depend on how close the experimental results are to the S-N curve. Figure 10a shows the high–low loading where the first maximum stress is denoted by $\sigma_{max1}$ and the second maximum stress is denoted by $\sigma_{max2}$. Each location of point *b* was found according to Equation (7) with a validated exponent *n* = 10.1 using the Matlab script in the Appendix A. Additionally, detailed numerical values are listed in Table 1 with accuracies

calculated using $(\log N_f - \log N_{f2})/\log N_f$. The prediction of the remaining fatigue lives appear to be in good agreement with experimental results.

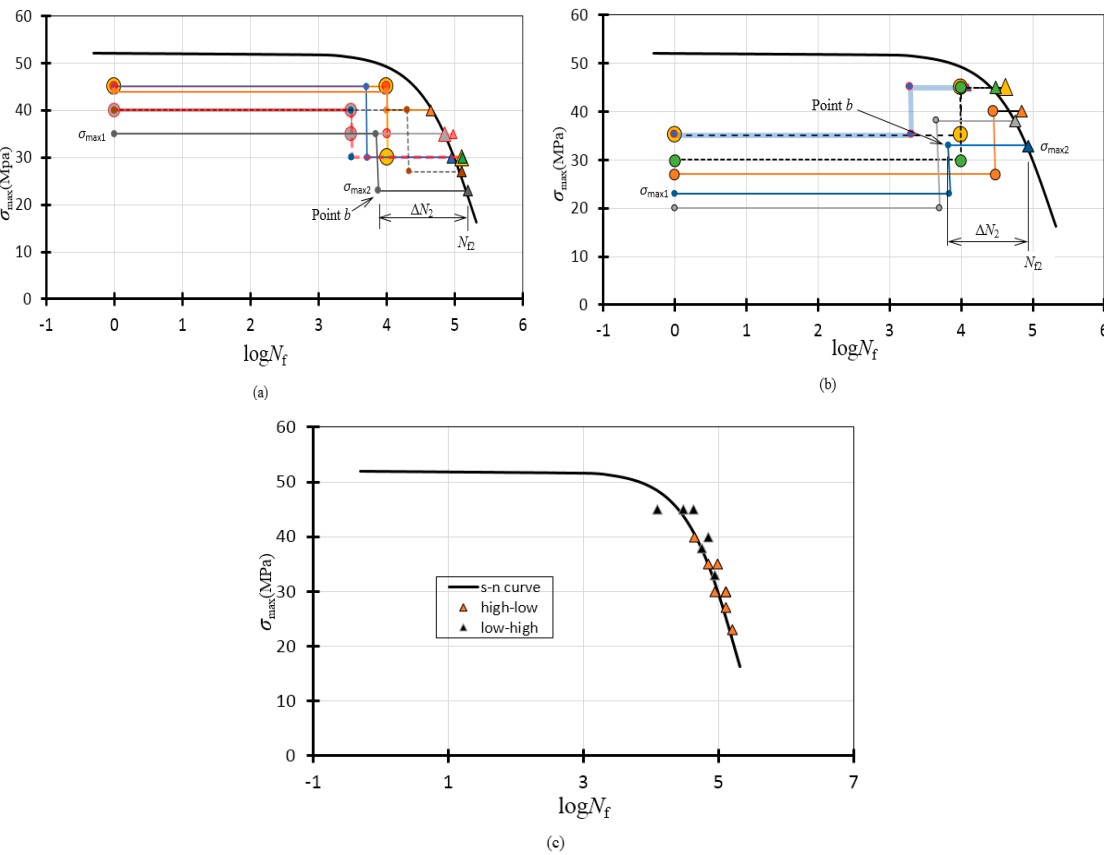

**Figure 10.** Remaining fatigue life prediction represented by an S-N curve and experimental data points represented by triangular data points: (**a**) high-low loading paths; (**b**) low-high loading paths; and (**c**) combined experimental results.

**Table 1.** High-low fatigue loading for remaining fatigue life.

| $\sigma_{max1}$ (MPa) | $\log N_1$ (Cycles) | $\sigma_{max2}$ (MPa) | At Point $b$ log (Cycles) | Experimental Remaining Loading Cycles at $\sigma_{max2}$ | $\Delta N_2$ | $N_{f2}$ | Accuracy (%) |
|---|---|---|---|---|---|---|---|
| 45 | 4.000 | 30 | 4.015 | 116,159 | 1.087 | 5.102 | 2.23 |
| 45 | 3.699 | 30 | 3.712 | 84,013 | 1.238 | 4.950 | −0.81 |
| 45 | 4.000 | 35 | 4.006 | 83,331 | 0.965 | 4.971 | 2.56 |
| 44 | 4.000 | 40 | 4.001 | 34,640 | 0.649 | 4.650 | −0.37 |
| 40 | 3.477 | 35 | 3.481 | 67,458 | 1.367 | 4.848 | 0.03 |
| 40 | 3.477 | 30 | 3.489 | 127,098 | 1.626 | 5.115 | 2.48 |
| 40 | 4.301 | 27 | 4.323 | 105,463 | 0.779 | 5.102 | 0.68 |
| 35 | 3.845 | 23 | 3.874 | 149,425 | 1.322 | 5.196 | 0.62 |

On the other hand, the low-high loading results are given in Figure 10b and Table 2. Additionally, a good agreement between predictions and experimental results is seen, although some scatter at $\sigma_{max2}$ = 45 MPa may be noticed. The scatter may be due to the instability from the high stress ($\sigma_{max2}$ = 45 MPa) which is close to the ultimate strength of 45.65 at the crosshead speed of 1 mm/min. A combined plot for both high-low and low–high loadings is given in Figure 10c, showing the clustered data points along the S-N curve as an overall view of the experimental data points.

**Table 2.** Low–high fatigue loading for the remaining fatigue life.

| $\sigma_{max1}$ (MPa) | $\log N_1$ (Cycles) | $\sigma_{max2}$ (MPa) | At Point $b$ log (Cycles) | Experimental Remaining Loading Cycles at $\sigma_{max2}$ | $\Delta N_2$ | $N_{f2}$ | Accuracy (%) |
|---|---|---|---|---|---|---|---|
| 35 | 4.000 | 45 | 3.994 | 31,978 | 0.628 | 4.622 | 4.85 |
| 35 | 3.301 | 45 | 3.296 | 10,440 | 0.798 | 4.094 | −7.11 |
| 30 | 4.000 | 45 | 3.986 | 20,251 | 0.490 | 4.476 | 1.56 |
| 27 | 4.477 | 40 | 4.454 | 42,731 | 0.398 | 4.852 | 3.97 |
| 20 | 3.699 | 38 | 3.657 | 53,186 | 1.104 | 4.761 | 0.34 |
| 23 | 3.845 | 33 | 3.820 | 80,589 | 1.120 | 4.940 | 0.68 |

## 6. Conclusions

S-N fatigue behaviour has been studied using PETG specimens manufactured with a 3D printer to conclude the following:

A new simple method for S-N curve characterisation is successfully developed to avoid the conventional erroneous way of collecting the data point at the lowest number of loading cycles.

The theoretical method for predicting the remaining fatigue life is verified at a stress ratio (*R*) of 0.4 (higher than zero) for the first time.

Fatigue damage consisting of cracks and whitening is described.

**Author Contributions:** Validation, H.S.K.; formal analysis, S.H.; investigation, H.S.K.; resources, S.H.; data curation, H.S.K.; writing—review and editing, H.S.K.; All authors have read and agreed to the published version of the manuscript.

**Funding:** This research received no external funding.

**Acknowledgments:** The authors would like to thank Michael Roberts of the School of Engineering for his technical assistance with the 3D printing of specimens and mechanical testing.

**Conflicts of Interest:** The authors declare no conflict of interest.

## Appendix A

Matlab script for determination of *α* and *β* for S–N curve, and exponent n

```
clear all
close all
N0=0.5% first cycle
% Experimental data:
Su=52 % Ultimate strength
logNf=[-0.3 3.73 3.80 4.09 4.17 4.25 4.43 4.49 4.63 4.83 4.99 5.11 5.27 4.74 4.91 5.20];%
Smax=[Su 50 49 48 47 46 44 43 40 35 30 25 20 38 33 23]; % Experimental results
n=length(logNf) % Number of experimental data points
N=input('*** Enter differentiation method (e.g. 5point or 7point) - default=3point ');
if isempty(N),
        N=3; % 3-point differentiation method
end
x=10.^logNf, y=1-Smax/Su
dy=diff(y); % for numerical differentiation:
dx=diff(x); % for numerical differentiation:
if N==3% 3-point differentiation method
        for i=1:(n-1-(N-1)/2);
                Avedy= (dy(i)+dy(i+1))/(N-1);
                dydx(i)=Avedy/((dx(i)+dx(i+1))/(N-1));
                  XXX=log10(Smax);
```

```
                        XX(i)=XXX(i+1);%for plot
                    end
        elseif N==5; % 5-point differentiation method
                for i=1:(n-2-(N-1)/2);
                    Avedy= (dy(i)+dy(i+1)+dy(i+2)+dy(i+3))/(N-1);
                    dydx(i)=Avedy/((dx(i)+dx(i+1)+dx(i+2)+dx(i+3))/(N-1));
                    XXX=log10(Smax);
                        XX(i)=XXX(i+2);
                end
            elseif N==7;% 7-point differentiation method
                for i=1:(n-3-(N-1)/2);
                    Avedy= (dy(i)+dy(i+1)+dy(i+2)+dy(i+3)+dy(i+4)+dy(i+5))/(N-1);
                        dydx(i)=Avedy/((dx(i)+dx(i+1)+dx(i+2)+dx(i+3)+dx(i+4)+dx(i+5))/(N-1));
                            XXX=log10(Smax);
                        XX(i)=XXX(i+3);
                end
        end
        figure, plot(XX,log10(dydx),'square','MarkerSize', 8,'MarkerEdgeColor', 'k', 'Marker-
FaceColor', [1 0 0])
        xlabel('log\sigma_m_a_x'),ylabel('log \DeltaD_f/\DeltaN_f ')
            p=polyfit(XX,log10(dydx),1);
            beta=p(1);
            alpha=10^p(2);
                hold on
                    xx=linspace(min(XX),max(XX),5);
                    yy=log10(alpha)+beta.*xx;
            plot(xx,yy, 'k-','Linewidth', 2),
            TTT=num2str(beta),
TTT2=num2str(alpha)
            title(['\alpha=' TTT2 ' \beta=' TTT])
        %% n (expo) value determination:
                            N=input('Have you set values for validation? If yes, press "Enter" ');
                            if isempty(N),
                            end
        expo=0.5 %initial exponent trial
        interval=1 %****interval between High and Low stresses
        Lowest=20 % lowest stress of range for validation
        gap=2 %***** difference (MPa) between Su and Smax for validation
        for i=1:inf
                j=0
                SHmax=Su-gap; % high (σ_Hmax) and low (σ_Lmax) stresses
                SLmax=SHmax-interval;
        while SLmax >= Lowest
                j=j+1;
        DfB=1-SHmax/Su;
        DfA=1-SLmax/Su;
        dfb=(DfB/DfA)^(1/expo);%dfb calculation
        logNfB=log10(((Su^(-beta))/(alpha*(beta-1)))*((SHmax/Su)^(1-beta)-1)+N0);
        logNfC=log10(((Su^(-beta))/(alpha*(beta-1)))*((SLmax/Su)^(1-beta)-1)+N0);
        dfC=(logNfB+0.3)/(logNfC+0.3);%dfC calculation

        Difff=dfb-dfC;
        Dif(j)=Difff;
                        SHmax=SLmax;
```

```
                                SLmax=SLmax-interval;
            end
                if min(Dif)>= 0
                        break
                else
                        expo=expo+0.1
                end
            end
            nValue=num2str(expo);
            % Plot S–N curve with data using alpha and beta
            hold off
            % plot data points:
            figure, plot(logNf, Smax, 'o','MarkerSize', 8,'MarkerEdgeColor', 'k', 'MarkerFace-
Color', [1 0 0])
            xlabel('Log N_f cycles'), ylabel('\sigma_m_a_x (MPa)')
                hold on
                x=linspace(logNf(1),logNf(n)+0.1,500);
                ySmax=Su*(alpha*(beta-1)*(10.^x-N0)./(Su.^(-beta))+1).^(1/(1-beta)); %tS–N
curve
                plot(x,ySmax,'k-','Linewidth', 2)
                ylim([0 Su+10])
                    title(['\alpha=' TTT2 ' \beta=' TTT ' n value=' nValue])
            legend('Experimental data points','Kim and Zhang S–N model')
```

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
