# Peer review of "S-N Curve Characterisation for Composite Materials and Prediction of Remaining Fatigue Life Using Damage Function"

_jcs, doi:10.3390/jcs5030076_

Round 1
Reviewer 1 Report
The title is much too general. Please specify at least the special range of application and perhaps also the methods used.
Abbreviations have to be explained (e.g. PETG in Abstract).
Author Response
- As suggested, “for composite materials” and “using damage function” are inserted in Title.
- As suggested, “(polyethylene terephthalate glycol-modified)” next to “PETG” is inserted in Abstract.
- Spelling is checked and corrected for ‘tempearture’ in section ‘3.2. Mechanical test’ on page 6. Note that the style is in British English.
Reviewer 2 Report
The manuscript is well-written and ready for publication with some minor corrections as below:
- Do the authors use R=0.5 or 0.4 for fatigue tests? In the abstract section, you write R=0.5. Please clarify.
- How many samples did you test to take into account the variation in fatigue life? Calculating the remaining fatigue life is challenging because of the scatter in the fatigue data and the amount of fatigue damage will likely to be varied.
- The damage mechanism of the fatigued samples should be described rather than reporting the failure appearance.
- The conclusions can be shown with annotation of the data.
Author Response
- The value ‘0.5’ in Abstract is changed to ‘0.4’.
- The numbers of samples are directly seen in Figure 8 for S-N curve, and Table 1, Table 2, and Figure 10c for load changes.
- The following is inserted in the last paragraph on page 9: “The damage mechanism of the fatigued specimens mainly involves crack initiation and then propagation. The observations here would represent the final stage of the mechanism involved.”
- The comment with ‘annotation’ does not make sense but “(R)” is inserted under ‘6. Conclusion’.